# Chirp-Based FHSS Receiver with Recursive Symbol Synchronization for Underwater Acoustic Communication

**DOI:** 10.3390/s18124498

**Published:** 2018-12-19

**Authors:** Geunhyeok Lee, Woongjin Park, Taewoong Kang, Kiman Kim, Wanjin Kim

**Affiliations:** 1Department of the Radio Communication Engineering, Korea Maritime and Ocean University, Busan 49112, Korea; bigstar0097@kmou.ac.kr (G.L.); woongjin3438@kmou.ac.kr (W.P.); taewoong@kmou.ac.kr (T.K.); 2Agency for Defense Development, Changwon 51678, Korea; rmsgur1313@nate.com

**Keywords:** covert underwater acoustic communication, frequency-hopping spread spectrum, symbol synchronization, Doppler fading, fractional Fourier transform, underwater channel simulator

## Abstract

In this paper, we propose a covert underwater acoustic communication method that is robust to fading using a chirp signal combined with a frequency-hopping spread spectrum scheme. A fractional Fourier transform, which estimates the slope of the signal frequency variation, is applied to the receiver to enable a robust and reliable symbol estimation with respect to the frequency and irregular phase variations. In addition, since the recursive symbol synchronization can be implemented using a chirp signal, compression and expansion effects due to the Doppler shift can be mitigated. Simulation and lake trials were performed to verify the performance of the proposed method. The simulation was performed by two different methods.

## 1. Introduction

The upcoming 5G communication standard has been an active area of research in recent years, focusing on terrestrial communications. However, progress in underwater communications has been comparatively much slower. The underwater acoustic environment is much more challenging, with far greater dispersion and multipath effects, limiting the potential use of the classical 5G scheme in the underwater domain [1]. Some research in developing underwater technology has taken place with a focus on orthogonal frequency division multiplexing (OFDM) and single-carrier frequency division multiple access (SC-FDMA) schemes [2,3]. In general, the underwater acoustic communication community has focused on increasing transmission distance, improving bandwidth efficiency, and reducing the rate of error in the complex time–space-varying underwater channel [4,5]. Covert underwater acoustic communication has been studied for military purposes to hide positioning information or data information. Although it is not an acoustic signal, underwater optical communication using a blue–green laser has also been aimed at enhancing covertness through the use of the strong directivity of lasers [6]. In addition, biomimetic signaling has been studied as a scheme for underwater acoustic communications, but the subject is in the early stages [7,8]. On the other hand, the spread spectrum schemes are widely used in the field to achieve covert communication. There are two typical spread spectrum schemes. One is a direct-sequence spread spectrum (DSSS) method in which the data are multiplied by a spread factor [9,10,11,12]. The other is a frequency-hopping spread spectrum (FHSS) method in which the frequency band is shifted using a spread code [13]. The FHSS method is randomly hopped according to the hopping pattern. By this method, an interceptor who does not know the hopping pattern cannot recover the signal. Furthermore, if any jamming is executed with malicious intent, the signal performance will be maintained because the center frequency will be continuously changed. This capability is called anti-jamming [14,15]. A simple multiple-access underwater acoustic communication protocol called JANUS was designed and tested as a NATO standard. It was a binary frequency shift keying (BFSK)-based FH method [16]. A covert underwater acoustic communication scheme between an unmanned underwater vehicle (UUV) and a distant mother platform using an OFDM had been studied [17]. However, the previous FHSS method was of the form in which a single center frequency was hopped. A chirp-based method utilizes a data-modulated signal, which has its energy spread over a bandwidth that is greater than the rate of information being sent, like all spread spectrum systems. The chirp-based method is one in which the binary information modulates the slope of a linear chirp, with chirps representing zeros or ones. This method alone can be considered a covert underwater acoustic communication technology. Due to the wide bandwidth in the chirp-based method, immunity against various frequency-selective fading and a frequency diversity effect may be expected [18,19].

Fractional Fourier transform (FrFT) is a generalized analysis method of the existing Fourier transform concept [20,21]. In particular, when a chirp signal has a linear characteristic in the time–frequency domain, robust detection is possible in a noisy or reverberant environment, as the FrFT results can obtain a concentrated energy spectrum. The FrFT has been applied in various fields, such as radar, image processing and communication [22,23,24,25]. Since underwater acoustic communication signals must typically pass through poor underwater channels, the coherent methods that are sensitive to phase shifts complicate the structure of the receiver. On the other hand, the non-coherent methods, such as the FrFT, can mitigate the fading effects with a relatively simple receiver structure.

The chirp signal combined with FHSS and demodulated by the FrFT method was studied in recent times [26]. This method simultaneously transmitted two chirp signals in one symbol duration and demodulated the received frequency-hopped signals in the FrFD (fractional Fourier domain) of the FrFT spectrum. As the up-slope chirp signal and the down-slope chirp signal were applied in one symbol duration, a parallel receiver should have retrieved the different transform orders.

In this paper, we propose the single chirp-based FHSS method demodulated by the non-coherent FrFT method. In the proposed method, either the up-slope chirp signal or the down-slope chirp signal is transmitted within one symbol duration, and does not require the parallel demodulator. There is also a difference from the conventional method in symbol demapping using the FrFT spectrum. The conventional method was independently demodulated using different orders for the up-slope chirp and the down-slope chirp without considering the negative frequency domain in the fractional frequency domain. On the other hand, since the proposed method uses the symmetrical characteristic of the frequency (having the positive frequency and negative frequency), there is a difference in the form represented by the FrFT due to the slope inversion of the positive and the negative frequency domains. In addition, as only a single transform order is required, a simpler receiver structure can be used.

Due to the movement, a Doppler shift can occur to the received signal, and the Doppler frequency will be relatively large due to the slow acoustic velocity in water compared to the propagation velocity on the ground [27]. Since the received signal is expanded and compressed by the Doppler shift, a method of implementing symbol synchronization recursively using an autocorrelation function of the estimated chirp signal should be employed. To consider the performance of the proposed method, we conducted simulations based on an underwater channel propagation model and conducted a lake trial to verify its validity.

The remainder of this paper is organized as follows: In the next section, the principle of the single chirp-based FHSS scheme is presented. Section 3 describes the receiver based on the FrFT. Section 4 and Section 5 then show the performance of the proposed method as evaluated via simulations and lake trials. Section 6 provides the conclusion.

## 2. Chirp-Based FHSS Scheme

The basic chirp signal used in the proposed method is expressed in Equation (1) as
(1)$$s\left(t\right)=\exp\left[j2\pi \left(f_{0}t+\frac{k}{2}t^{2}\right)\right],$$
where $f_{0}$ is the initial frequency and k is the chirp rate. Equation (1) can be rewritten to express the chirp signal as follows,
(2)$$s_{u}\left(t\right)=\exp\left[j2\pi t\left(f_{min}+\frac{f_{max}-f_{min}}{T}t\right)\right],$$
(3)$$s_{d}\left(t\right)=\exp\left[j2\pi t\left(f_{max}+\frac{f_{min}-f_{max}}{T}t\right)\right],$$
where $f_{min}$ is the minimum frequency and $f_{max}$ is the maximum frequency. T is the duration of the symbol. Equation (2) represents a signal that has an up-slope frequency with an initial frequency $f_{min}$. Equation (3) represents a signal that has a down-slope frequency with an initial frequency $f_{max}$. To express the chirp signal in real numbers, the above equations can be expressed as
(4)$$S_{u}\left(t\right)=Re\left\{s_{u}\left(t\right)\right\},$$
(5)$$S_{d}\left(t\right)=Re\left\{s_{d}\left(t\right)\right\}.$$

The chirp-based scheme selects one of the chirp symbols of Equation (4) using the binary sequence $b_{i}$. Therefore, the chirp-based signal $s_{c}\left(t\right)$ with the symbol period *T* is expressed as [28]
(6)$$s_{c}\left(t\right)=\{\begin{array}{c} S_{u}\left(t\right),\;if\;b_{i}=0\\ S_{d}\left(t\right),\;if\;b_{i}=1 \end{array},\;nT\leq t\leq \left(n+1\right)T.$$

This equation implies assigning a binary value “1” to a signal with an up-slope frequency, and assigning a binary value “0” to a signal with a down-slope frequency. During the duration of the symbol, the chirp signal is mapped by the binary bit stream. The received signal is usually used by a correlator receiver.

Figure 1 shows the basic structure of the hopping scheme in a chirp-based FHSS. In this case, since the hopping duration is longer than the symbol duration, there are several symbols in the single hopping interval.

## 3. Receiver Design Based on Fractional Fourier Transform

Figure 2 shows a block diagram of the proposed method. Firstly, a hopping pattern is generated and multiplied by the modulated transmission signal to perform frequency hopping, and the receiver utilizes a bandpass filter with a known hopping pattern. Then, the compression or expansion effect of the signal due to the Doppler shift can be corrected by introducing the matched filter results between the symbol estimation through the FrFT spectrum and the received signal.

In the example shown in Figure 1, there are four symbols in a single hopping frequency interval. Each symbol is divided into a low-frequency band and a high-frequency band. The symbol with an up-slope in the low-frequency band is called “Symbol Mapping #1”, and the symbol with a down-slope is called “Symbol Mapping #2”. The symbol with up-slope in the high-frequency band is called “Symbol Mapping #3”, and the symbol with down-slope is called “Symbol Mapping #4”.

Figure 3a shows the matched filter result, which is from the auto-correlation function of the “Symbol Mapping #1” in any hopping pattern. It can be seen that the output of the matched filter has a maximum value at the ideal point. Figure 3b shows the matched filter result, which is the cross-correlation function between the “Symbol Mapping #1” and “Symbol Mapping #2” in any hopping pattern. Since “Symbol Mapping #1” and “Symbol Mapping #2” have different slopes, the maximum point of the matched filter result cannot be correctly represented, and the magnitude is less than the ideal value [29].

Figure 4a shows the matched filter result from the cross-correlation function between “Symbol Mapping #1” and “Symbol Mapping #3” in any hopping pattern. The mapping types have the same slopes but show a lower magnitude because the frequency bands are different. Figure 4b shows the matched filter result as a cross-correlation function between “Symbol Mapping #1” and “Symbol Mapping #4” in any hopping pattern. The mapping types have the same frequency, but show lower values due to their different slopes.

In this paper, we propose a method to estimate the symbol in the received signal by applying the FrFT scheme. The FrFT is a generalization of the conventional Fourier transform concept, which makes it possible to convert the time domain signal into a time–frequency intermediate domain. That is, when the time domain is represented by the *x*-axis and the frequency domain by the *y*-axis, the FrFT is the result obtained by rotating an arbitrary angle ∅ in the counterclockwise direction. The FrFT can obtain the spectra with various characteristics depending on the transform orders defined as α = 2∅/π. If the different signals have different frequency slopes, the signals will have peaks at different positions with different transform orders on the FrFT spectrum. In this way, the correct transform order with peaks in each FrFT spectrum is called the optimal transform order, which is denoted by $\alpha_{opt}$ [27,30]. The FrFT equation is
(7)$$F_{a}f\left(u\right)=\int K_{\alpha }\left(u,x\right)f\left(x\right)dx,$$
where the term $K_{\alpha }\left(u,x\right)$ is expressed as
(8)$$K_{\alpha }\left(u,x\right)=\{\begin{array}{c} A_{\alpha }\exp\left\{i\pi \left[\cot\left(\alpha \right)\left(x^{2}+u^{2}\right)-2\csc\left(\alpha \right)ux\right]\right\},\;\left(\alpha \neq n\pi \right)\\ \delta \left(u-x\right)\;\left(\alpha =2n\pi \right)\\ \delta \left(u+x\right)\;\left(\alpha =\left(2n\pm 1\right)\pi \right) \end{array},$$
where $A_{\alpha }$ is the amplitude factor defined as $\sqrt{1-icot\left(\alpha \right)}$. If ∅=π/2, the FrFT is equivalent to the Fourier transform. Next, the optimal transform order in a discrete region can be expressed as
(9)$$\alpha_{opt}=\frac{2}{\pi }\emptyset =\frac{2}{\pi }cot^{-1}\left(\frac{Mk}{f_{s}{}^{2}}\right),$$
where $f_{s}$ is the sampling frequency, k is the chirp rate, and M represents the sampling point. If the sampling points and sampling frequency are fixed, the optimal transform order is determined from the value k.

Figure 5a shows a conceptual diagram of the FrFT for a chirp signal. In this case, we can assume that the chirp signal $x_{u}\left(t\right)$ with the up-slope characteristic exists for the analysis period T, and the chirp signal $x_{u}\left(t\right)$ has the initial frequency $f_{0}$ at the start time (kτ−∆T/2) and the frequency $f_{1}$ at the end time (kτ+∆T/2) (generally, the frequency $f_{1}$ is higher than the frequency $f_{0}$). Therefore, the point $o_{u}$ which is the middle point between $p_{u}=\left[\left(k\tau -∆T/2\right)/s,\;f_{0}s\right]$ and point $q_{u}=\left[\left(k\tau +∆T/2\right)/s,\;f_{1}s\right]$ can be expressed as [$k\tau /s,\;f_{m}s$], where $f_{m}$ is the center frequency and s is the rescaling factor, and is defined as $s=T/f_{s}$. The k is the symbol number. The vector $u_{0}$, which is the point $o_{u}$ that corresponds to the u-axis, is
(10)$$u_{0}=\left|o_{u}\right|\sin\left(\alpha_{opt}\frac{\pi }{2}\right)=\left[f_{m}s-\frac{k\tau }{s}\right]\sin\left(\alpha_{opt}\frac{\pi }{2}\right).$$

When the starting point for the signal $x_{u}\left(t\right)$ is τ=0, the vector $u_{0}$ can be written as:(11)$$u_{0}=\left(f_{m}s\right)\sin\left(\alpha_{opt}\frac{\pi }{2}\right).$$

When considered in the frequency domain, the signal also has a negative frequency that is inversely proportional to the positive frequency. The negative frequency exists symmetrically with the positive frequency and forms a relatively wide and low spectrum on the FrFT as [$p_{u}^{\prime }\;to\;q_{u}^{\prime }$] in Figure 5a. On the other hand, in Figure 5b, the chirp signal $x_{d}\left(t\right)$ with the down-slope characteristic will appear symmetrically with the signal $x_{u}\left(t\right)$. Therefore, the vector $u_{1}$ can be written as
(12)$$u_{1}=\left(-f_{m}^{\prime }s\right)\sin\left(\alpha_{opt}\frac{\pi }{2}\right),$$
where $f_{m}^{\prime }$ is the center frequency of the signal $x_{d}\left(t\right)$. If the signal $x_{u}\left(t\right)$ and $x_{d}\left(t\right)$ are in the same frequency band and have the same slope, the vector $u_{1}$ can be rewritten as
(13)$$u_{1}=\left(-f_{m}s\right)\sin\left(\alpha_{opt}\frac{\pi }{2}\right)=-u_{0}.$$

The opposite characteristics of FrFT can be seen from Equation (13). If the center frequency and the frequency variation are same, the positions of the vectors $u_{0}$ and $u_{1}$ are the opposite and the same value.

Figure 6a shows the symmetrical characteristics in the fractional domain to distinguish between the up-slope and the down-slope. This is an example of the FrFT spectrum for Figure 5a. In the proposed method, with respect to the middle line of the fractional spectrum, the left side is the positive frequency domain, and the right side is the negative frequency domain (theoretically existing). Therefore, in the up-slope state (like Figure 5a), the energies are concentrated on the left side, and in the down-slope state (like Figure 5b), the energies are concentrated on the right side. On the other hand, Figure 6b shows the FrFT spectrum of the conventional method that coexists with the up-slope and the down-slope in the symbol duration. Unlike the proposed method, the conventional method is processed only in the positive frequency domain and distinguishes between the up-slope and the down-slope by using the two transform orders which are dependent on each symbol. This means that the conventional receiver must have the parallel structures. In this paper, we propose the FrFT receiver using only one transform order.

In the proposed method, when the frequency-hopping interval is $f_{wd}$, the vector $u_{hn}$ regarding the hopped frequency $f_{hn}$ is written as
(14)$$f_{hn}=f_{m}+nf_{wd}.$$

Then,
(15)$$\begin{array}{l} \;\;\;\;u_{hn}=\left(f_{hn}s\right)\sin\left(\alpha_{opt}\frac{\pi }{2}\right)\\ =\left[\left(f_{m}+nf_{wd}\right)s\right]\sin\left(\alpha_{opt}\frac{\pi }{2}\right)\\ =u_{0}+\left(nf_{wd}s\right)\sin\left(\alpha_{opt}\frac{\pi }{2}\right), \end{array}$$
where n represents the number of hopped intervals. The signals hop due to $f_{wd}$, which means that they are spaced at regular intervals in the FrFT spectrum. Using the above property, regardless of the frequency band of the hopping, a ‘decision section’ for estimating a symbol can be formed because the FrFT bin formed for each symbol is always regular. That is, the FrFT bin is obtained for the received signal through Equation (15), and the symbol is estimated by the demapping method which distinguishes the FrFT bin in the decision section. Since the estimated symbol can be implemented as an ideal form of the signal, it is possible to perform a matched filter between the received signal and the ideal signal. The synchronization error can be obtained between the results of the above procedure and the results of the matched filter in the ideal case. When compensating for the next symbol along with the weight, it is possible to implement symbol synchronization to correct the errors associated with the Doppler shift.

Figure 7 is a plot of the FrFT spectrum for all frequency-hopping bands in a single space. From the relation given in Equation (15), it can be seen that the spectrum is formed regularly in proportion to the frequency-hopping interval. Based on this, even if an arbitrary frequency hopping is performed, it will be completed in the above spectrum section. That is, in the case that any hopped symbol is input, it can be estimated through the section of the spectrum.

## 4. Simulation and Results

We performed the simulation to analyze the performance of the proposed method. Figure 8 shows a packet structure of a transmission signal designed for the simulation. We used 1080 bits per frame, a center frequency of 16 kHz, a sampling frequency of 192 kHz, and a data rate of 100 bps. One chirp symbol band is 1.2 kHz and the total band is approximately 10 kHz. The LFM (linear frequency modulation) signals are used to distinguish the packets. The signal length is 0.1 s and the bandwidth is 4 kHz. A guard interval of 1 s is arranged to prevent an interference due to the signal. The pseudo-noise code uses 512 bits for 0.512 s, and no channel coding technique is applied.

There are several software packages for simulating channel transfer characteristics in underwater environments. Norwegian FFI provides the Watermark software that simulates transmission information in several underwater channels [31]. However, to verify the performance of the proposed method, a channel is simulated by applying the sound velocity distribution measured in the South Sea, South Korea, to the VirTEX (Virtual Time Series Experiment), which is a Bellhop-based underwater channel modeling program developed and released by the Scripps Institution of Oceanography (SIO) [32,33]. The simulation environment is shown in Figure 9. In the simulation, it is assumed that the sea floor is inclined at a depth of 30 m from the transmitter to 40 m from the receiver. The position of the receiver is placed at a depth of 30 m and the transmitter is assumed to be moving horizontally with a fixed depth of 5 m. It is assumed that the distance between the transmitter and receiver moves from 300 m to 200 m linearly at approximately 12 knots towards the receiver and then moves in the opposite direction at the same speed from 200 m to 300 m.

The channel response characteristics for the simulation are shown in Figure 10. The channel impulse response shows that not only the direct path, but also the sub-path, are reflected from the surface and the floor.

Figure 11a shows the result of the matched filter for the termination by overlapping the received signal and the ideal signal when approaching between the transmitter and the receiver. As shown in Figure 11a, the signal is compressed since it is positioned backward, even though the packet synchronization is initially adjusted due to the Doppler frequency, which is caused by the mobility of the transmitter. The synchronization error due to the compression is approximately 0.056 s, and assuming a transmission rate of 100 bps, the error is approximately 5–6 symbols.

On the other hand, Figure 11b shows an overlapping of the received signal and the ideal signal in the case of retreating, which expands the signal due to the Doppler frequency. The synchronization error due to the expansion is approximately 0.055 s and has an error of approximately 5 to 6 symbols, as in the previous case.

Table 1 shows the bit error rate according to the signal-to-noise ratio (SNR) in the presence or absence of the symbol synchronization. The symbol length changes continuously due to the compression and expansion of the signal. Accordingly, if the synchronization is not performed, a large error occurs. On the other hand, when the symbol synchronization is performed using the proposed method, the uncoded bit error rate (BER) is considerably low. The simulation results show that the proposed method is robust to multipath delays and variability due to the Doppler shift.

The second simulation was performed using the underwater channel software released by FFI. Watermark is a benchmark for physical-layer schemes of underwater acoustic communications. It contains data with five different channel characteristics, and we used a channel (named as NCS1) measured in Norway. This is the channel in which broad Doppler spreading occurs and repeats for 60 cycles. We set the Doppler shift higher to make more visible the performance difference of the proposed method. Other detailed information is provided in the reference [31]. Figure 12 shows the bit error rate. The horizontal axis of the figure refers to the number of cycles. The bit error rate of the proposed method is approximately 0.07, which shows better performance than the conventional method.

## 5. Lake Trials

To investigate the performance of the proposed method in a real environment, lake trials were conducted at Kyungcheon Lake, Mungyeong City. The first and second experiments were conducted at the beginning and end of May 2018, respectively. The water depth of the lake was approximately 30–40 m, and the transmitters were located approximately 5 m from the boat. The receiver was located at a depth of 30 m, and the boat was steered randomly from 100 m to approximately 400 m from the receiver. The boat moved at speeds of 2 knots and 4 knots, and the configuration of the signals used was the same as those of the previous simulations. However, in the second experiment, which was conducted at the end of May, the hopping rate designed a half-spacing signal for each hopping frequency. Because of the overlap of the hopping frequencies, the ISI can degrade the communication performance due to multipath delay. However, this can be overcome by using the FrFT and bandpass filtering. Additionally, by designing with the above method, the frequency band can be used more efficiently. A Neptune D/17/BB model projector was used as the transmitter and a B&K 8106 hydrophone was used as the receiver. The Neptune D/17/BB model that was used as a transmitter is a broadband projector with a bandwidth of approximately 10 kHz [34,35].

Figure 13 shows the channel impulse response and Figure 14 shows the scattering function used for the Doppler estimation. Figure 13a shows the measured channel response in the first experiment performed at the beginning of May, and Figure 13b is the result of the second experiment performed at the end of May. As shown in Figure 13, the difference in the reception time occurs as the transmitter moves, which generates a slope. Figure 13b shows there are more complex delay profiles. Figure 14a shows that the Doppler frequency is approximately −16 Hz to −22 Hz, and Figure 14b shows that the Doppler frequency is approximately −10 Hz to −5 Hz, where the (−) sign means that the distance between the transceivers gradually increased. However, there seems to be an issue with the scattering function in Figure 14b, as a main Doppler shift cannot be detected and the multipath arrival is entirely smeared. This was a result of the increased level of noise due to minor equipment problems and because the channel conditions in the second experiment were relatively poor due to stronger winds.

Figure 15a and Figure 16a show spectrograms of the transmitted signal, and Figure 15b and Figure 16b show spectrograms of the received signal through the lake experiments. From the figures, we can see that there is considerable fading in the received signals due to multipath propagation delay.

Since the received signal experiences substantial fading as it passes through the underwater channel, the delay due to multipath propagation or the slope change due to the Doppler frequency should be considered. Therefore, it is necessary to calculate the range adjacent to the optimal transform order for the application of the FrFT scheme at the receiver. Figure 17 shows the accumulation of the spectra of the received signal that is obtained from the optimal transform order to the adjacent range. When the optimal transform order is $\alpha_{opt}=0.996$, the spectrum is spread by the delay caused by multipath propagation in the range adjacent to the order. However, due to the property of the chirp signal, the multipath has little effect on the estimation. Figure 17a shows the correctly estimated symbol, which is on the right side in the FrFT bin and fixed on the number 240 of the FrFT bin point. We can determine from this symbol that the down-slope chirp signal has a center frequency of 1.2 kHz. On the other hand, Figure 17b shows the incorrectly estimated symbol. The receiver estimates to a different point. The transmitted signal has the transform order $\alpha_{opt}=0.996$, but the received signal has the approximate value “1”. This means that the receiver cannot distinguish between left and right. Therefore, we determined this symbol as the correct center frequency and the incorrect slope. In this case, the received symbol was affected by the serious Doppler effect beyond the Doppler tolerance of the chirp signal.

Figure 18 shows the cumulative error rate per elapsed symbol to analyze the synchronizing effect of the proposed method. If the symbol synchronization is not performed, the cumulative errors increase because the symbol is compressed or expanded over time. Conversely, the performance of the synchronized symbol is good. As a result, the FrFT receiver can be a system used to efficiently demodulate chirp signals that are robust and resistant to fading and Doppler effects.

Table 2 and Table 3 show the results of the uncoded bit error rate obtained by demodulating the received signal from the lake experiment, where the transmitted signal is not applied to the channel coding. Table 2 and Table 3 show the cases where the moving speed of the transmitter is approximately 2 knots and 4 knots, respectively. Some experiments were performed in an environment where the weather condition was relatively poor, and the corresponding performance may be lower than those of other cases. Nevertheless, according to the proposed method, the bit error rate is reduced to 36.4% after iterative symbol synchronization. Therefore, we verified that the proposed method improves the performance. The chirp signal has properties that are robust to fading and Doppler effects, and the proposed method showed that it was possible to maintain these advantages through efficient demodulation. To compare the computations, we compared the computation times for each of the arbitrary symbols designed with the proposed method and the conventional method. The parameters of symbols were set to be similar to those shown in Figure 6. In this case, by using the MATLAB function “Tic − Toc”, the running program time was measured and compared. The proposed method used a single receiver and scanned the value around the transform order to obtain the spectrum (see Figure 17). On the other hand, the conventional method used at least two receivers, which means that two transform orders must be calculated. As a result of a relative computation time comparison with a computer equipped with an Intel Core i7-6700/4 GHz processor, the proposed method took 0.2277 s while the conventional method took 0.4717 s.

## 6. Conclusions

In this paper, we proposed a covert acoustic communication scheme that is robust to fading and Doppler effects. It consists of a single chirp-based FHSS that is resistant to delay propagation and the Doppler shift. Additionally, the receiver demodulates to the FrFT spectrum. To verify the performance of the proposed method, we estimated the signal in an environment with a multipath delay, and the Doppler frequency through simulations and lake experiments, to determine the uncoded BER. Since the chirp signal has a relatively good orthogonality, it is robust against the instability of the phase. It was also possible to estimate the slope using the FrFT spectrum, even in the case of frequency variation due to the Doppler shift. This means that the proposed method can be demodulated while maintaining the properties of the chirp signal. As a result, we could obtain high-reliability communication performance. In the conventional method, the up-sloped chirp signal and the down-sloped chirp signal existed within one symbol simultaneously, and independent demodulators with different orders were used. However, in the proposed method, either the up-sloped or down-sloped chirp signal was transmitted within one symbol, and a parallel structure, as used with the conventional method, was not required in the demodulator. To verify the performance of the proposed method, we performed various simulations. Specifically, we used the Watermark software released by FFI. In the future, it will be necessary to conduct research for efficient FrFT spectrum formation and optimization of the receiver structure. Sea trials will be performed.

## Figures and Tables

**Figure 1 sensors-18-04498-f001:**
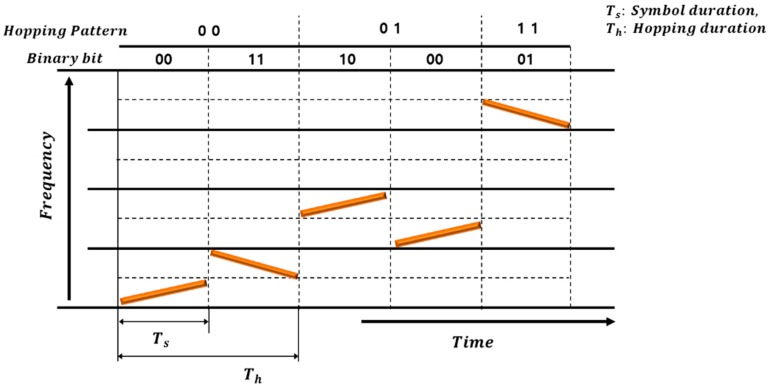
Basic scheme of the chirp-based FHSS.

**Figure 2 sensors-18-04498-f002:**
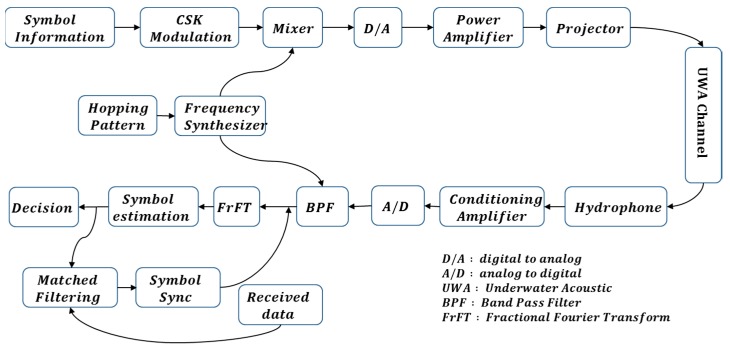
Block diagram of the proposed method.

**Figure 3 sensors-18-04498-f003:**
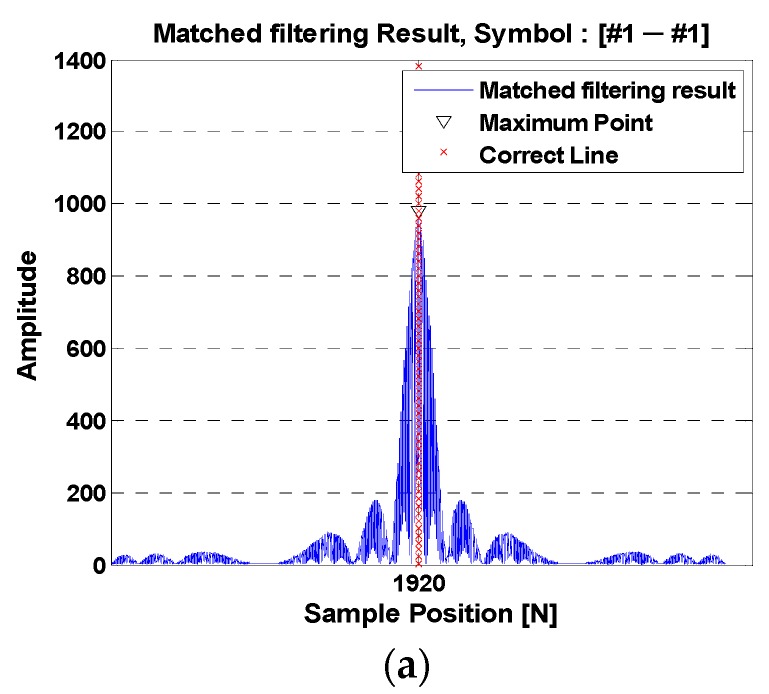
Matched filter output in the same frequency band. (**a**) The same slope symbol; (**b**) different slope symbols.

**Figure 4 sensors-18-04498-f004:**
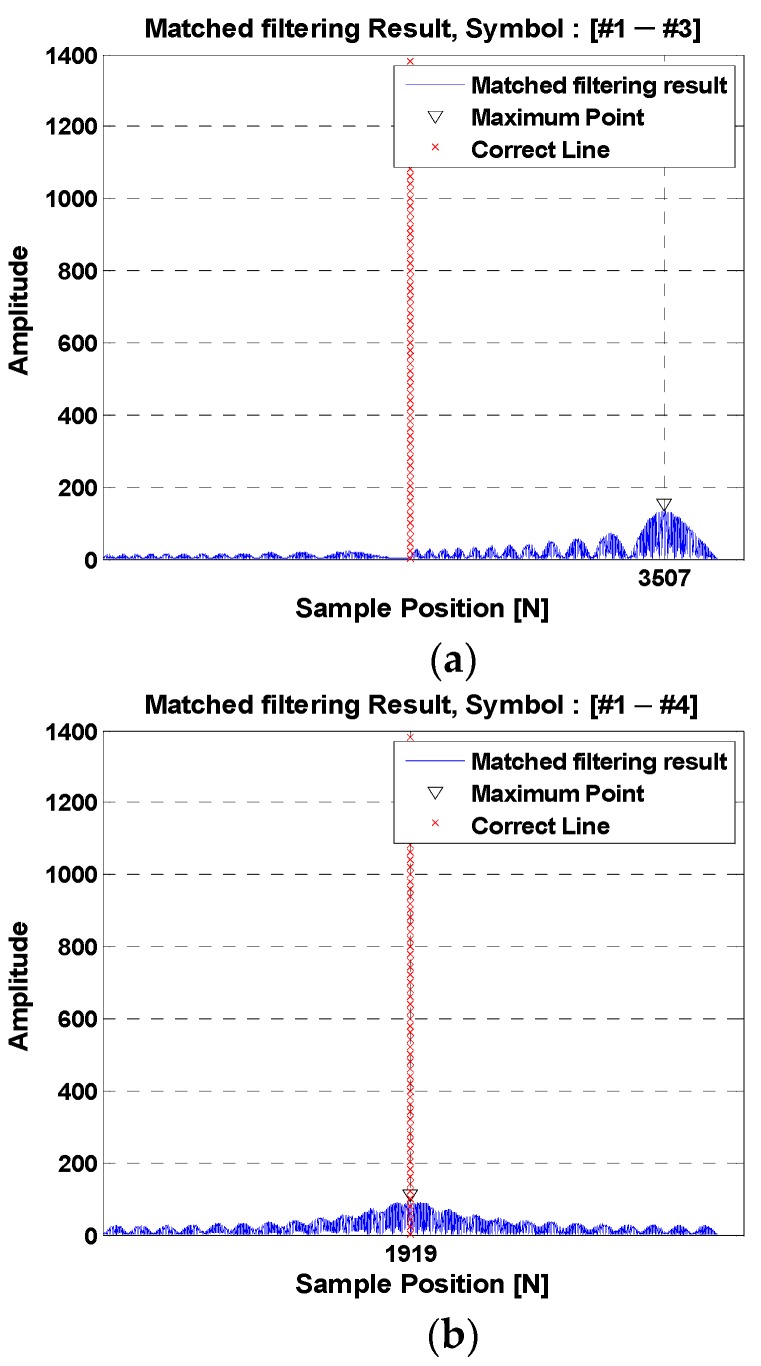
Matched filter output in different frequency bands. (**a**) The same slope symbol; (**b**) different slope symbols.

**Figure 5 sensors-18-04498-f005:**
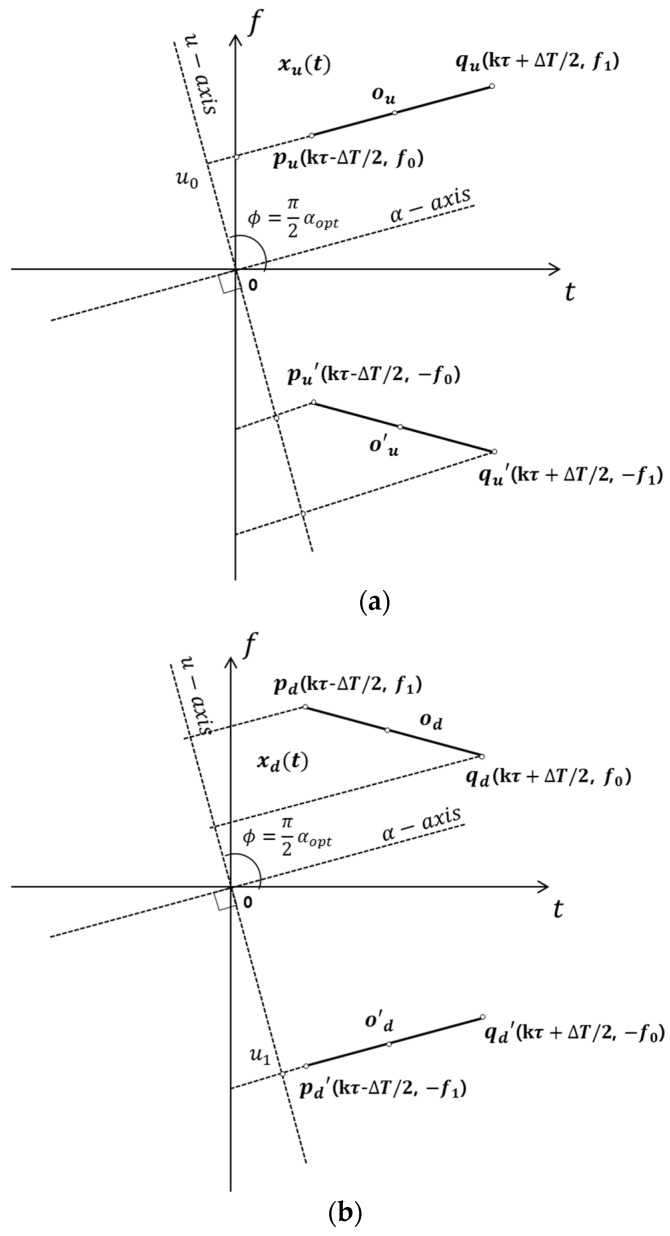
Relation between the chirp signal and FrFT coefficient α. (**a**) The up-slope chirp signal; (**b**) the down-slope chirp signal.

**Figure 6 sensors-18-04498-f006:**
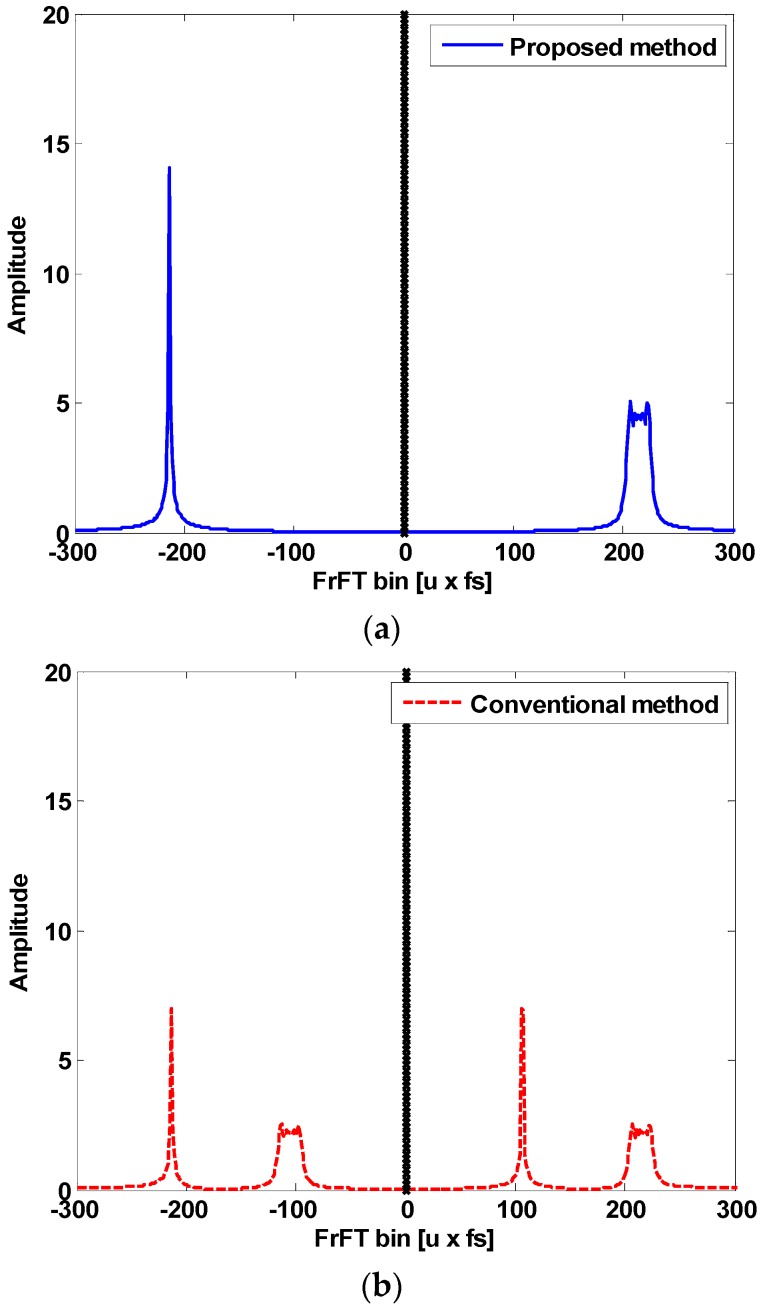
FrFT spectrum comparison between (**a**) the proposed method, and (**b**) the conventional method.

**Figure 7 sensors-18-04498-f007:**
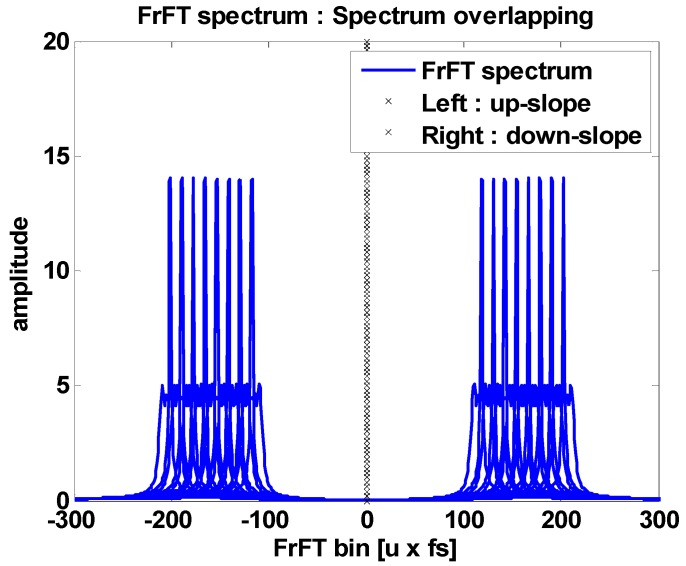
Aggregate FrFT spectra for all frequency bins.

**Figure 8 sensors-18-04498-f008:**
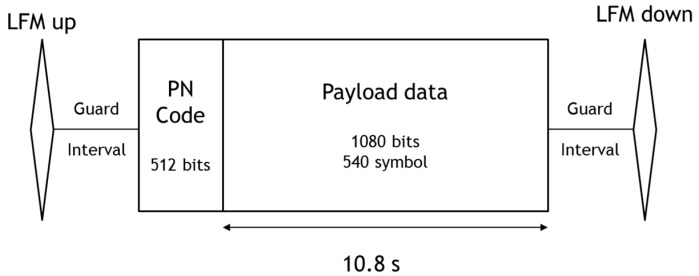
Packet design.

**Figure 9 sensors-18-04498-f009:**
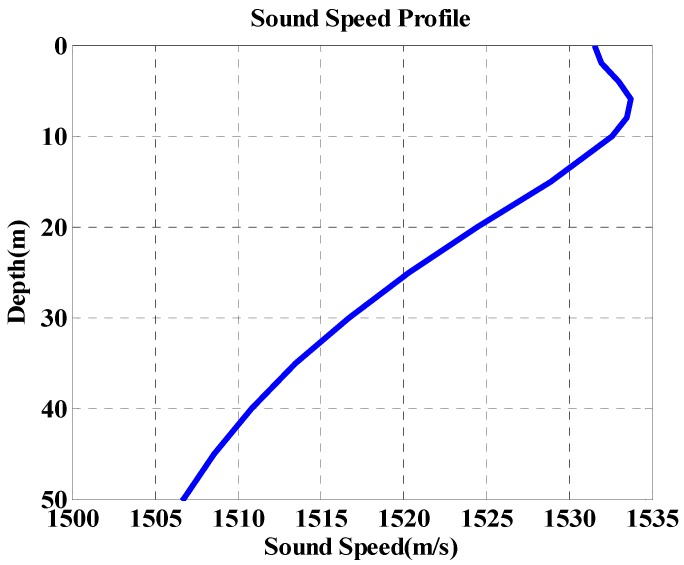
Sound velocity profile for the simulation.

**Figure 10 sensors-18-04498-f010:**
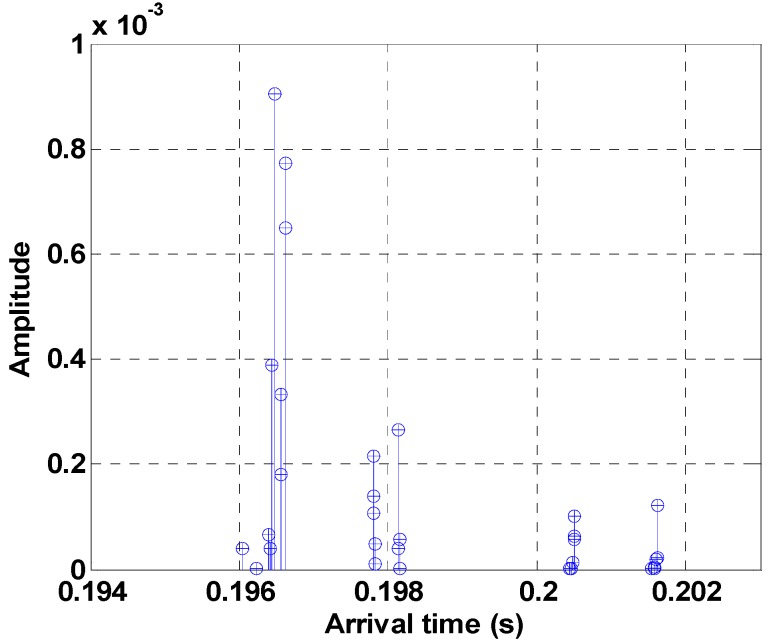
Channel impulse response for simulation.

**Figure 11 sensors-18-04498-f011:**
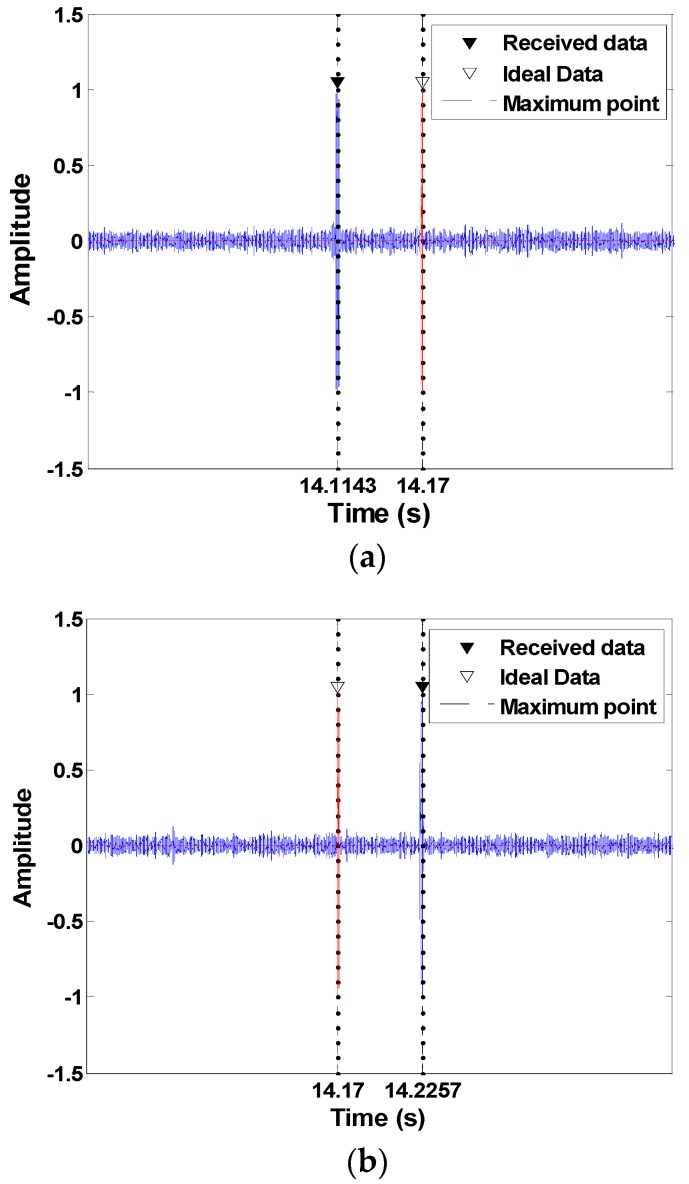
Matched filtering output for the received data and the ideal data. (**a**) The approach; (**b**) the retreat.

**Figure 12 sensors-18-04498-f012:**
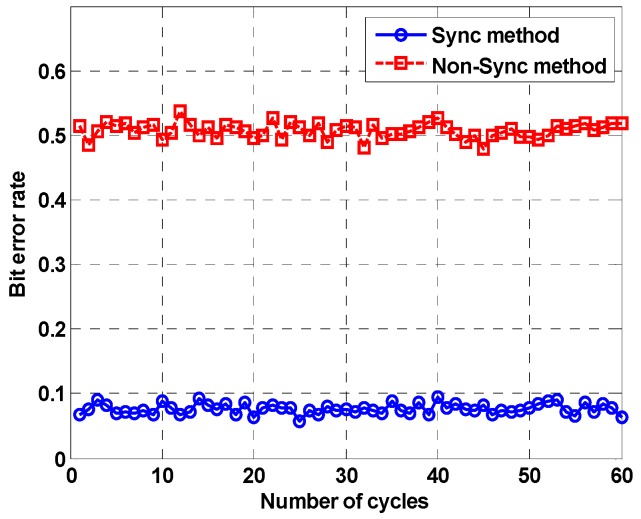
Simulation result using Watermark benchmark model.

**Figure 13 sensors-18-04498-f013:**
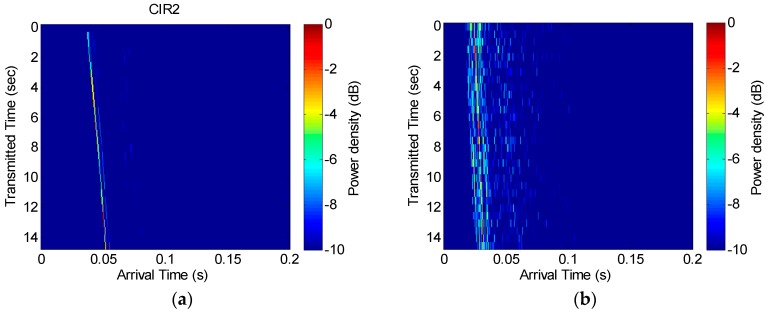
Measured channel impulse responses for (**a**) the first lake trial and (**b**) the second lake trial.

**Figure 14 sensors-18-04498-f014:**
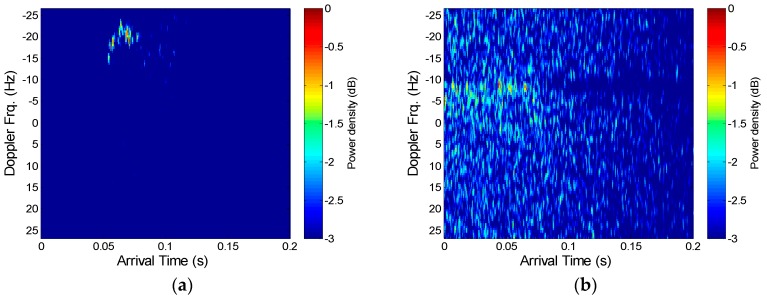
Scattering function for the Doppler frequency estimation for (**a**) the first lake trial and (**b**) the second lake trial.

**Figure 15 sensors-18-04498-f015:**
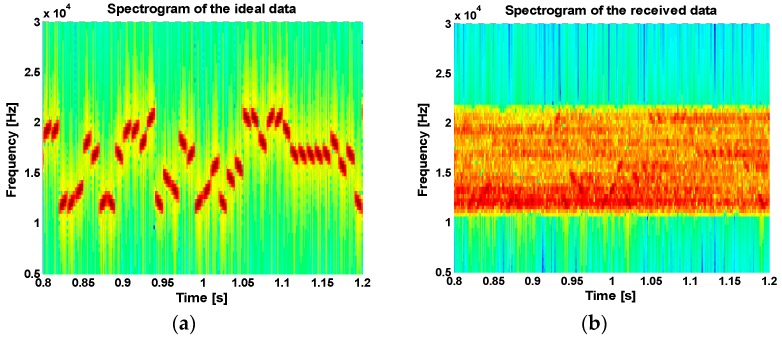
Spectrogram of (**a**) the transmitted signal, and (**b**) the received signal in the first trial.

**Figure 16 sensors-18-04498-f016:**
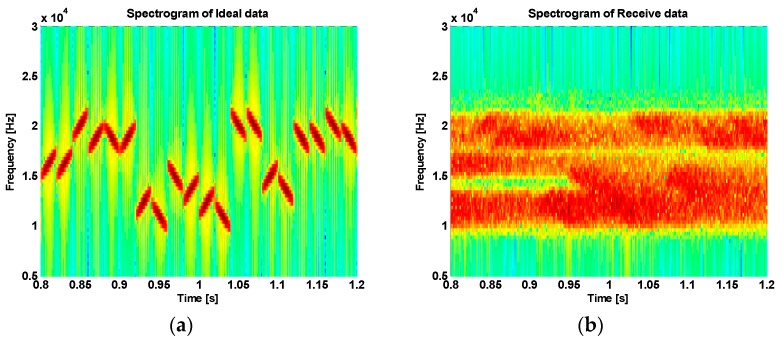
Spectrogram of (**a**) the transmitted signal, and (**b**) the received signal in the second trial.

**Figure 17 sensors-18-04498-f017:**
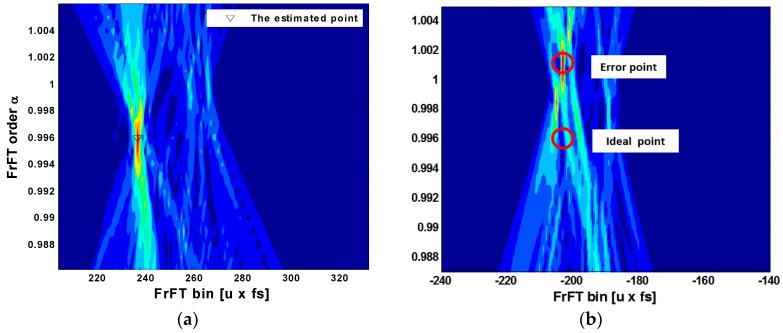
Color intensity of the FrFT results in (**a**) the correctly estimated symbol and (**b**) the incorrectly estimated symbol.

**Figure 18 sensors-18-04498-f018:**
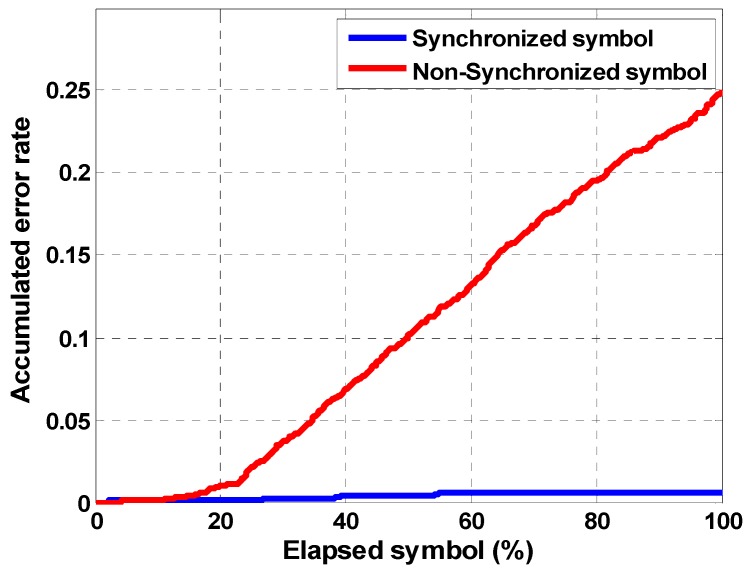
Synchronization performance.

**Table 1 sensors-18-04498-t001:** Uncoded bit error rate in the simulation.

SNR (dB)	Non-Sync Method	Sync Method
15	0.49	0.000
10	0.48	0.002
5	0.49	0.004
0	0.48	0.050

**Table 2 sensors-18-04498-t002:** Uncoded bit error rate at 2 knots.

Trial	First Trial		Second Trial	
	Non-Sync Method	Sync Method	Non-Sync Method	Sync Method
1	0.23	0.09	0.19	0.08
2	0.24	0.09	0.35	0.11
3	0.28	0.001	0.10	0.09
4	0.12	0.09	0.08	0.07
5	0.22	0.11	0.10	0.01
6	0.09	0.01	0.07	0.06

**Table 3 sensors-18-04498-t003:** Uncoded bit error rate at 4 knots.

Trial	Non-Sync Method	Sync Method
1	0.17	0.04
2	0.26	0.05

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
