# Peer review of "Chirp-Based FHSS Receiver with Recursive Symbol Synchronization for Underwater Acoustic Communication"

_sensors, 2018, doi:10.3390/s18124498_

Reviewer 1 Report

This manuscript discusses how to improve error performance of frequency hopping communication system with chirp signal by using FrFT.

However, I cannot understand the difference between the method proposed in this manuscript and the conventional method of using FrFT[1]. The existing method has been described up to the line 223. And then it is claimed that the proposed method is different from existing method with figure 6. However, the FrFT spectrum shown in figure 6 is shown as the difference between transmitting only 1 and 2 symbols for fixed interval. There is no explanation as to how the proposed method is technically different. The conventional method mentioned in this manuscript seems to indicate only a technique that does not repeatedly synchronize. But repetitive synchronization is a method applied to all modern communication systems. Therefore, this manuscript is highly demanded for originality and novelty.

In conclusion, I cannot recommend the publication of this manuscript in this stage.

[1] Xingbin Tu, Xiaomei Xu, Zheguang Zou, Liangliang Yang, and Jianming Wu, “Fractional Fourier domain hopped communication method based on chirp modulation for underwater acoustic channels,” Journal of Systems Engineering and Electronics, vol.28, no.3, 2017.

Author Response

Response to reviewer’s comments

Reviewer 1’s comments :

This manuscript discusses how to improve error performance of frequency hopping communication system with chirp signal by using FrFT. However, I cannot understand the difference between the method proposed in this manuscript and the conventional method of using FrFT[1]. The existing method has been described up to the line 223. And then it is claimed that the proposed method is different from existing method with figure 6. However, (1) the FrFT spectrum shown in figure 6 is shown as the difference between transmitting only 1 and 2 symbols for fixed interval. There is no explanation as to how the proposed method is technically different. (2) The conventional method mentioned in this manuscript seems to indicate only a technique that does not repeatedly synchronize. But repetitive synchronization is a method applied to all modern communication systems. Therefore, this manuscript is highly demanded for originality and novelty.

Author’s Response :

We thank the reviewer for the suggestion. We have now added the following text in section 1 to incorporate the suggestion :

-          We tried to express the difference between the two methods to Figure 6, but we admit that it was failed. So Figure 5 and equation [10]~[13] are revised to explain the proposed method in more detail. Then, in relation to Figure 6, we describe the conventional method and the proposed method. Two methods use the similar scheme, but there are clearly differences. The important differences are the number of transform order and the scope of analysis. Multiple transform orders of conventional method mean that the receiver operates in parallel, and the negative frequency domain is not considered. On the other hand, the proposed method uses only a single transform order, both the positive frequency domain and the negative frequency domain are meaningful. The details are in the paper.

-          We understood about your second comment. We wanted to express that it is possible to demodulate the received signal using the property of the chirp signal even in environments with multipath fading and Doppler shift, during the symbol was synchronizing. We have corrected the factors that could lead to misunderstanding. We have also added the error problem about the proposed method in Figure 17b to help the understanding of our manuscript.

Thank you for your valuable comments of our paper, once again.

Reviewer 2 Report

This is the second time this paper is reviewed.  I do recommend this paper to be published for its interesting technical content.  However, I do suggest to the authors to apply a thorough review of the English.

Fig 5 could be improved.  Could we show points p, q, and u1? 

The z-axis does not have any units in Figure 14.  There seems to be an issue with the scattering function in Fig 14b, since a main Doppler shift cannot be detected, and that the multipath arrival is entirely smeared.  Intuitively, this seems wrong. 

Author Response

Reviewer 2’s comments :

This is the second time this paper is reviewed.  I do recommend this paper to be published for its interesting technical content.  However, I do suggest to the authors to apply a thorough review of the English.

(1) Fig 5 could be improved. Could we show points p, q, and u1? (2) The z-axis does not have any units in Figure 14. (3) There seems to be an issue with the scattering function in Fig 14b, since a main Doppler shift cannot be detected, and that the multipath arrival is entirely smeared. Intuitively, this seems wrong.

Author’s Response :

We thank the reviewer for the suggestion. We have now added the following text in section 1 to incorporate the suggestion :

-        Thanks to your suggestion, our paper is more refined. Regarding Figure 5 and some equations, we realized that the explanation was not enough. So, we revised the Figure 5 and Equation [10]~[13] to help explaining the proposed method.

-        We added the unit of z-axis to compare the power density between the environment and the received signal in Figure 13 and Figure 14. The reason for using these figures is to show the environment in which the signal is affected by the multipath delay and Doppler shift.  

-        The main Doppler shift can be shown in Figure 14 which is the scattering function. We reduced the power density to show the Doppler effects more clearly in Figure 14. In Figure 14b, we think that it was difficult to detect the multi-path delay and Doppler shift because some minor problems had occurred with the equipment used at the time. However, if you look closely, you can see that Doppler shift had been occurring in -5 ~ -10 Hz. It is reflected in the paper.

Thank you for your valuable comments of our paper, once again.

Reviewer 3 Report

The authors focus their study on underwater acoustic communication and they propose a method that is robust to fading using a chirp signal combined with a frequency-hopping spread spectrum scheme.

The manuscript is overall well-written and easy to follow. This manuscript seems to be a revision of a previous submission, thus, it seems that parts of the paper are rewritten. The proposed analysis is concrete and coherent.

The authors should address some minor comments to improve their manuscript. Initially, the provided literature review is misleading as multiple references are referred together, e.g. 1-3, 7-13, as if the cited papers are doing exactly the same thing. The latter is not true and the authors should provide the main concepts of each cited paper, while many of them are redundant.

Moreover, the authors should highlight that the current 5g communication, of which the underwater communications are part of, adopt the OFDMA technique for the downlink communication and the SC-FDMA for the uplink communication ("A survey on uplink resource allocation in OFDMA wireless networks." IEEE Communications Surveys & Tutorials 14, no. 2 (2012): 322-337, "Uplink resource allocation in SC-FDMA wireless networks: A survey and taxonomy." Computer Networks 96 (2016): 1-28.), where the proposed framework can be applied to both types of communication.

Also, the authors should explain the derivation of equation 6. How the chirp-based signal is defined and how this consists one of the authors’ contributions? To the best of the reviewer’s knowledge (please justify if I am wrong), this is already defined in the literature even from the 3G underwater communication adopting older types of multiple access methods. Figure 1 needs to be enlarged as it is not readable. The reviewer cannot see the legend and the content of the figure (extremely small). How the proposed scheme can improve the energy-efficiency of the communication which is one major challenge in 5G communication (see: "Energy-efficient subcarrier allocation in SC-FDMA wireless networks based on multilateral model of bargaining." In IFIP Networking Conference, 2013, pp. 1-9. IEEE, 2013, "Underwater acoustic sensor networks: research challenges." Ad hoc networks 3, no. 3 (2005): 257-279, "Underwater acoustic networks–issues and solutions." International journal of intelligent control and systems 13, no. 3 (2008): 152-161.)? What is the implementation complexity of the proposed framework? (justify with some indicative numerical results). Overall, this is a very interesting paper that should be slightly improved in order to be better motivated and positioned within the existing literature. The manuscript needs a minor revision and a detailed check for minor typos, grammar and syntax errors.

Author Response

Reviewer 3’s comments :

The authors focus their study on underwater acoustic communication and they propose a method that is robust to fading using a chirp signal combined with a frequency-hopping spread spectrum scheme.

The manuscript is overall well-written and easy to follow. This manuscript seems to be a revision of a previous submission, thus, it seems that parts of the paper are rewritten. The proposed analysis is concrete and coherent.

The authors should address some minor comments to improve their manuscript. Initially, the provided literature review is misleading as multiple references are referred together, e.g. 1-3, 7-13, as if the cited papers are doing exactly the same thing. The latter is not true and the authors should provide the main concepts of each cited paper, while many of them are redundant.

Moreover, the authors should highlight that the current 5g communication, of which the underwater communications are part of, adopt the OFDMA technique for the downlink communication and the SC-FDMA for the uplink communication ("A survey on uplink resource allocation in OFDMA wireless networks." IEEE Communications Surveys & Tutorials 14, no. 2 (2012): 322-337, "Uplink resource allocation in SC-FDMA wireless networks: A survey and taxonomy." Computer Networks 96 (2016): 1-28.), where the proposed framework can be applied to both types of communication.

Also, the authors should explain the derivation of equation 6. How the chirp-based signal is defined and how this consists one of the authors’ contributions? To the best of the reviewer’s knowledge (please justify if I am wrong), this is already defined in the literature even from the 3G underwater communication adopting older types of multiple access methods. Figure 1 needs to be enlarged as it is not readable. The reviewer cannot see the legend and the content of the figure (extremely small). How the proposed scheme can improve the energy-efficiency of the communication which is one major challenge in 5G communication (see: "Energy-efficient subcarrier allocation in SC-FDMA wireless networks based on multilateral model of bargaining." In IFIP Networking Conference, 2013, pp. 1-9. IEEE, 2013, "Underwater acoustic sensor networks: research challenges." Ad hoc networks 3, no. 3 (2005): 257-279, "Underwater acoustic networks–issues and solutions." International journal of intelligent control and systems 13, no. 3 (2008): 152-161.)? What is the implementation complexity of the proposed framework? (justify with some indicative numerical results). Overall, this is a very interesting paper that should be slightly improved in order to be better motivated and positioned within the existing literature. The manuscript needs a minor revision and a detailed check for minor typos, grammar and syntax errors.

Author’s Response :

We thank the reviewer for the suggestion. We have now added the following text in section 1 to incorporate the suggestion :

-        Thank you for your suggestion. We revised the paper a little in the results and conclusions to address our minor comments. And, the provided literature review is misleading as multiple references are referred together. So, we divided the reference properly to exclude misleading.

-         

-        Through the first paragraph of the paper, we have highlighted the 5G communication of underwater acoustic which is adopted the OFDM and SC-FDMA, and adds the reference about reviewer's comments.

-         

-        In the case of Equation 6, it is a generally known symbol mapping method. Equation 6 expresses how Equation 4 or Equation 5 is selected depending on the state of the bit stream. For Figure 1, we reshaped the figure to increase the resolution and to help understanding the proposed method. The reader may understand easier to Equation 6 through the reshaped Figure 1. We also added references related to Equation 6.

-         

-        The energy efficiency we mentioned previously was an opinion based on a relative comparison with the conventional methods. The conventional method transmitted signals in two bands at any time, but this paper referred to energy efficiency because it used only one band. By the comment of the reviewer, the readers might misunderstand, so we omitted this part.

-         

-        To compare computational loads, we had compared the computation times for each of the arbitrary symbols designed with the proposed method and the conventional method. In this case, by using the MATLAB function called "Tic - Toc", the running program time had been measured and compared. The results are shown in the manuscript.

Thank you for your valuable comments of our paper, once again.

This manuscript is a resubmission of an earlier submission. The following is a list of the peer review reports and author responses from that submission.

Round  1

Reviewer 1 Report

This manuscript discusses how to improve error performance of frequency hopping communication system with chirp signal by using FrFT and synchronizing repeatedly in the receiver.    

However, the method to improve the performance using the FrFT in frequency hopping communication system based on the chirp modulation for underwater acoustic channels has already been introduced by Xingbin Tu[1]. There are many similar papers that deal with communications using FrFT [2-5]. In addition, the method of correcting the timing offset repeatedly has been already well known and has been adopted in various communication standards. The method that authors insist they are presenting newly in this manuscript is not different from the existing studies, and therefore, I could not find out any insight of this paper in the manuscript. The test authors conduced in lake is worthy of note, but only the presenting of the result of communication data analysis is not sufficient for me to recommend the publication of this paper if there is no any improvement of method or proposal of new method.      

I think it would be good if you investigate the performance variation due to the underwater channel condition with additional experiments    

Unfortunately, I cannot recommend this paper for publication at the present stage.

[1] Xingbin Tu, Xiaomei Xu, Zheguang Zou, Liangliang Yang, and Jianming Wu, “Fractional Fourier domain hopped communication method based on chirp modulation for underwater acoustic channels,” Journal of Systems Engineering and Electronics, vol.28, no.3, 2017.

[2] Fei Yuan, Qian Wei, and En Cheng, “Multiuser chirp modulation for underwater acoustic channel based on VTRM,” International Journal of Naval Architecture and Ocean Engineering, vol.9, 2017.

[3] Xin QiuXue-jun ShaQing-hua Shen, and Xuan-li Wu, “Fractional Fourier transform based secure communication system.” 6th International Conference on Wireless Communications Networking and Mobile Computing, 2010.

[4] Yixin Chen, Carmine Clemente, John J. Soraghan, and Stephan Weiss, “Partial fractional Fourier Transform (PFRFT)-OFDM for underwater acoustic communication,” 23rd European Signal Processing Conference, 2015.

[5]  Domenico Gaglione, Carmine Clemente, Christos V. Ilioudis, Adriano Rosario Persico, Ian K. Proudler, and John J. Soraghan, “Fractional Fourier based waveform for a joint radar-communication system,”  2016 IEEE Radar Conference, 2016.

Reviewer 2 Report

CW-based FH-SS is a standard, please see STANAG 4748 (please use the keyword ANEP 87 JANUS), The definition is e.g. here

https://www.researchgate.net/publication/265594031_The_JANUS_underwater_communications_standard

Covert underwater communication based on Chips was analysed also in the big European project UCAC 2006-2010.

https://www.researchgate.net/publication/228584312_UUV_COVERT_ACOUSTIC_COMMUNICATIONS

Use the realistic FFI-BENCHMARK  https://www.ffi.no/en/research-projects/Sider/Watermark.aspx to validate your approach - it is important to compare your results with others in the same environment.